# Weakly-Supervised Recommended Traversable Area Segmentation Using Automatically Labeled Images for Autonomous Driving in Pedestrian Environment with No Edges

**DOI:** 10.3390/s21020437

**Published:** 2021-01-09

**Authors:** Yuya Onozuka, Ryosuke Matsumi, Motoki Shino

**Affiliations:** Department of Human and Engineered Environmental Studies, Graduate School of Frontier Sciences, The University of Tokyo, 5-1-5, Kashiwanoha, Kashiwa, Chiba 277-8563, Japan; rmatsumi@edu.k.u-tokyo.ac.jp (R.M.); motoki@k.u-tokyo.ac.jp (M.S.)

**Keywords:** recommended traversable area detection, weakly-supervised semantic segmentation, pedestrian environment, autonomous driving

## Abstract

Detection of traversable areas is essential to navigation of autonomous personal mobility systems in unknown pedestrian environments. However, traffic rules may recommend or require driving in specified areas, such as sidewalks, in environments where roadways and sidewalks coexist. Therefore, it is necessary for such autonomous mobility systems to estimate the areas that are mechanically traversable and recommended by traffic rules and to navigate based on this estimation. In this paper, we propose a method for weakly-supervised recommended traversable area segmentation in environments with no edges using automatically labeled images based on paths selected by humans. This approach is based on the idea that a human-selected driving path more accurately reflects both mechanical traversability and human understanding of traffic rules and visual information. In addition, we propose a data augmentation method and a loss weighting method for detecting the appropriate recommended traversable area from a single human-selected path. Evaluation of the results showed that the proposed learning methods are effective for recommended traversable area detection and found that weakly-supervised semantic segmentation using human-selected path information is useful for recommended area detection in environments with no edges.

## 1. Introduction

In recent years, Mobility on Demand (MoD) services to deploy diverse mobility solutions for different mobility needs have attracted attention [1], aiming at providing last-mile transportation in environments where public transportation is insufficient, as well as to increase accessibility of social spaces and revitalize economic activities by increasing urban mobility. Among these services, the realization of autonomous personal mobility systems is expected [1,2].

Many autonomous mobility systems have been proposed for pedestrian environments such as sidewalks and community roads, among which many require accurate localization based on map information constructed using sensors such as a laser range finder (LRF). However, maintaining maps with precise structural information for all roads involves significant time and expense. In contrast, a method that uses a topological map for autonomous driving in unknown outdoor environments has also been proposed [3,4,5,6]. A topological map is a map represented by a combination of nodes on a road network representation, with links representing road sections between nodes. Existing systems using topological maps are generally based on road-following navigation using road boundaries, which are detected by edge features extracted from camera images [3,4] or physical features extracted by LRF [5,6]. However, when road-following navigation is applied in an environment where roadways and sidewalks coexist, as shown in Figure 1, a target point may be incorrectly set on a roadway based on mechanical traversability and continuity of the road surface texture. Personal mobility systems should avoid entering roadways because, if they enter and run along the road, they are likely to remain there. Therefore, in an environment where the roadway and sidewalk coexist, it is necessary to estimate a driving area that is both mechanically traversable and recommended by traffic rules and to navigate based on this estimation. In this paper, the area where the terrain is mechanically traversable and driving is recommended by traffic rules is defined as the recommended traversable area, and the degree of the recommendation is defined as driving recommendation degree.

In recent years, research on deep learning-based semantic segmentation, which can utilize human knowledge as training data, has been actively conducted for road scene segmentation [7,8]. Barnes et al. [9] proposed a method for detecting traversable paths based on automatically labeled data using recorded data from a data collection vehicle driven by a human driver. Human driving experience is considered to be based on human visual perception and prior knowledge such as traffic rules and can be useful for training semantic segmentation models in ambiguous environments with no edges.

In this study, we propose a recommended traversable area detection system that can be adapted to environments with no edges using automatically labeled data based on human knowledge and experience. The overall navigation system proposed is shown in Figure 2. This system consists of three planning stages: global path planning, local path planning, and motion planning. The global path is planned based on a topological map, while the local path is planned based on a map representing driving recommendation degree information, and the motion planning stage determines system operations, such as vehicle velocity and a steering angle, that appropriately compensate for safety based on spatial information obtained from the LRF. The role of the recommended traversable area detection system in the overall navigation system is to provide appropriate choices of driving directions when planning local paths. Appropriate choices mean that the system can detect multiple recommended directions when there is more than one recommended direction and that the system does not interpret areas where driving is not recommended as being suitable for driving, such as roadways where there is a distinction between roadway and sidewalk. The scope of this study is framed in red in Figure 2, and the main contributions in that part are as follows:An automatic image labeling method based on human knowledge and experience is proposed, as well as leaning methods to detect appropriate directions as recommended traversable areas. In particular, a data augmentation method and a loss weighting method are focused on as learning methods.An evaluation dataset is created and metrics are designed to evaluate the effectiveness of the proposed learning methods. Specifically, the recommended traversable area detection performance is evaluated in environments where roadways and sidewalks coexist.

The remainder of this paper is organized as follows. Section 2 provides a brief review of deep learning-based approaches for road scene segmentation. Section 3 describes an automatic image labeling method, learning methods, and an evaluation method. Section 4 presents the experiments, results, and discussion and demonstrates the effectiveness of the proposed learning methods. Finally, the conclusions of this study are presented in Section 5.

## 2. Related Works

In this study, a vision sensor was used for road scene segmentation in environments where it is difficult to evaluate the traversability based on physical features alone. Several vision sensor-based traversable area segmentation methods are based on region growing [10,11]. In region growing methods, traversable areas are expanded by sequentially evaluating the similarity of the color space and texture of neighboring pixels or superpixels from the seed pixel or superpixel assumed to be traversable. Since such similarity-based segmentation methods only evaluate the similarity between the pixel of interest and its surrounding pixels, it is likely to segment the roadway as a traversable area due to the continuity and similarity of the road surface in an environment such as that shown in Figure 1, which is undesirable in terms of traffic rules.

Recently, deep learning-based methods have been considered for road scene understanding benefiting from recent advances in deep learning. Meyer et al. [12] proposed a lane semantics detection method for autonomous driving without a highly accurate map. In this method, an ego-lane was detected even on a road without a centerline by adding image data distinguishing ego-lanes, parallel lanes, and opposite lanes to the CityScapes dataset [7] for training. In this way, even if a road cannot be segmented by focusing on pixel-level features, it can be segmented by capturing the relationships between pixels of the whole image using deep learning. However, deep learning requires a large dataset, the creation of which is time-consuming and labor-intensive. Although it is possible to use existing datasets, most of them are from the roadway viewpoint, and there is no large-scale dataset from the sidewalk viewpoint. In deep learning-based semantic segmentation, the effect of viewpoint change is significant, and we confirmed that the sidewalk on the image from the sidewalk viewpoint is classified as a roadway. Therefore, deep learning-based semantic segmentation in a pedestrian environment is not likely to be able to recognize the environment correctly, even if we use existing datasets. For the problem of creating datasets, methods for automatic labeling of drivable areas using a disparity map obtained from stereo images have been proposed [13,14]. However, the disparity map can only evaluate the mechanical traversability and cannot take traffic rules into account during automatic labeling. Barnes et al. proposed an automatic image labeling method using human driving experience. It involves information about mechanical traversability and a human-selected path, which can be useful in representing recommended traversable areas in a pedestrian environment. However, it has been pointed out as a problem that only a limited area of the width of the data collection vehicle can be output, and that it is not possible to detect the entire traversable area [12,15,16]. In contrast, Gao et al. [15] proposed a method to train separate neural networks to infer drivable and obstacle zones and to probabilistically represent whether a zone with ambiguous traversability is closer in attribute to a drivable zone or an obstacle zone. It was confirmed that this method can extract a wider drivable zone compared to the method of Barnes et al. [9]. Tang et al. [16] proposed a method to extend traversable areas by reflecting the results of multiple runs in training data using a global map created by LRF to represent the vehicle location in a global coordinate system. However, this method requires much effort to generate a training dataset, as it requires the preparation of a global map and multiple runs on the same road environment. In this paper, we propose learning methods that can detect an appropriate recommended traversable area from a single driving path for a single scene, allowing efficient generation of a training dataset.

## 3. Methods

### 3.1. Automatic Image Labeling Method

In this section, we describe an automatic image labeling method for semantic segmentation and a training dataset created using this method. In this study, the driving environment is classified into three classes: recommended area, non-traversable area, and other areas (traversable areas). In labeling, the area where a human has driven in general is defined as the recommended area because it contains information about mechanical traversability and human visual perception and prior knowledge such as traffic rules, and the area that cannot be physically travelled is defined as non-traversable area.

Following the pipeline shown in Figure 3, the automatically labeled dataset is used for offline training, and the trained model is used for online driving recommendation degree inference.

#### 3.1.1. Automatic Recommended Area Labeling

To label the area where a human has driven, we consider a projection of the three-dimensional (3D) position of the grounding point of the front wheels at time t+k onto an image $I_{t}$ at time *t*. Here, the 3D position of the grounding point is the position in the camera coordinate system $C_{t}$ at time *t* and is represented as${\;}^{C_{t}}X_{t+k}={\left[X,Y,Z\right]}^{T}$. The homogeneous coordinates of${\;}^{C_{t}}X_{t+k}$ are denoted by${\;}^{C_{t}}{\tilde{X}}_{t+k}={\left[X,Y,Z,1\right]}^{T}$. The projection point m=[u,v] on the image shown in Figure 4a is calculated using Equation (Equation 1), where $P\in R^{3\times 4}$ is the perspective projection matrix, $\tilde{m}={\left[u,v,1\right]}^{T}$ is the homogeneous coordinates of m, and *s* is a scale factor determined so that the third element on the left side of Equation (Equation 1) is 1.
(1)$$\begin{array}{c} s\tilde{m}=P\;{\;}^{C_{t}}{\tilde{X}}_{t+k} \end{array}$$

${\;}^{C_{t}}X_{t+k}$ is calculated using Equation (Equation 2) from the relationship between camera position and wheel grounding position, as shown in Figure 4. In Equation (Equation 2), $O_{0}$, $O_{t}$, and $O_{t+k}$ are the initial coordinates of a vehicle, the coordinates after *t* seconds, and the coordinates after t+k seconds, respectively, calculated from vehicle odometry. $T\in R^{4\times 4}$ represents the coordinate transformation matrix.${\;}_{O_{t}}^{C_{t}}T$ and${\;}^{O_{t+k}}X_{t+k}$ are derived from vehicle specifications, and${\;}_{O_{0}}^{O_{t}}T$ and ${\;}_{O_{t+k}}^{O_{0}}T$ are derived from vehicle odometry.
(2)$$\begin{array}{c} {\;}^{C_{t}}X_{t+k}={\;}_{O_{t}}^{C_{t}}T\;{\;}_{O_{0}}^{O_{t}}T\;{\;}_{O_{t+k}}^{O_{0}}T\;{\;}^{O_{t+k}}X_{t+k} \end{array}$$

For odometry, ORB-SLAM2 [17] with a visual stereo camera is adopted, which is a 3D odometry system independent of a vehicle model. When using visual odometry, if feature points on the image are insufficient and self-position information is lost, wheel odometry is used to interpolate.

The image $I_{t}$ at time *t* is labeled with a travelled area by filling in a rectangle consisting of the projected points of the left and right front wheels at a given time and their projected points at a time before the unit time. The projection ends when the wheel position reaches a point outside the camera’s field of view or at a distance greater than the threshold distance from the point where the projection started. This threshold value was set to 20 m from the maximum depth of the stereo camera (Stereolabs, ZED 2). It was confirmed in our previous study [18] that automatic recommended area labeling using this method can be performed with sufficient accuracy.

#### 3.1.2. Automatic Non-Traversable Area Labeling

It is undesirable to set the driving direction through the non-traversable area, and we describe an automatic non-traversable area labeling method. In this study, we used a method of object-based labeling using pre-trained models of semantic segmentation, similar to the work of Zhou et al. [19].

In this study, objects that are non-traversable are those that a vehicle may collide with, such as walls, cars, and pedestrians. We used the CityScapes dataset, where those objects are classified in the road scenes. Because the CityScapes dataset is from the roadway viewpoint, an autonomous mobility system trained on this model used to make driving path direction inferences on images from the sidewalk viewpoint would have a high probability of mistakenly recognizing a sidewalk as a roadway, due to the similarity of features and positional relationships of the images. On the other hand, as for the identification of objects that a vehicle may collide with, the possibility of misrecognition due to differences in viewpoints is low. This is because the features of an object are rarely similar to those of other objects and the objects exist in various locations, so the possibility of over-fitting for a certain location is low. In this study, we used training data remapped from 11 of 30 classes in the CityScapes dataset: road, wall, pole, terrain, person, vehicle, building, fence, vegetation, sky, and rider. When labeling a non-traversable area, it is undesirable to mistakenly label a traversable area as a non-traversable area because a local path cannot be planned in that area, leading to a less optimal path overall. Therefore, we adopted PSPNet [20] as our training network architecture. PSPNet has the ability to capture the global context through global average pooling in its pyramid pooling module, which acts to reduce misclassification of objects. Using this method, we expect to reduce the misclassification of road areas as non-traversable areas.

The learning conditions are shown in Table 1. However, the “poly” learning rate policy was used for the learning rate (lr), similar to the learning condition of Zhao et al. [20]. In addition, a pre-trained version of ResNet-50 [21], available in PyTorch [22], was used as an encoder. The dataset consisted of 4500 training data, including training and test data available in the CityScapes dataset, and the remaining 500 were used as validation data. In addition, a random mirror process was added as data augmentation and the model with the highest mean IoU was used for labeling.

Figure 5 shows examples of labeling using a model trained by PSPNet. As shown in the second column of Figure 5, there are areas that are not correctly labeled. For example, in Figure 5(b-2), there is an area labeled as a vehicle (blue) in the upper left part of the image, but, if you check the original image (b-1), you can see that the area is not actually a vehicle. However, since it is labeled as a non-traversable object, the model seems to have no problem in labeling them together as non-traversable, as shown in the third column of Figure 5. The fourth column of Figure 5 is an image of the third column of labeled non-traversable areas after labeling the recommended traversable areas using the method described in Section 3.1.1. In the case of a left or right turn, as shown in Figure 5(b-4), the recommended traversable area may overlap with a non-traversable area due to objects such as bicycles in front of the path, and such area is labeled as a non-traversable area. In addition, an example of labeling a traversable braille block as non-traversable is shown in Figure 5c. This result can lead to mistakenly estimating the traversable area as non-traversable area, which can affect path planning. Thus, it is necessary to understand how the training data affects the learning outcome. Therefore, we evaluate in Section 4 how well the trained model can predict non-traversable areas based on analysis of the training results.

#### 3.1.3. Dataset for Training

To generate training data using the automatic labeling method described in Section 3.1, we collected time-series images and vehicle state data (velocity and steer angle) from the experimental vehicle shown in Figure 6 driven by a human in a typical Japanese pedestrian environment, as shown in Figure 7. The collected data were then used to generate images labeled as recommended, non-traversable, and unclassified areas for every 0.5 m of driving distance. Table 2 shows the number of labeled images after doubling by horizontal flipping. In addition, the labeled images were sorted according to lateral displacement at the end of the trajectory projection and driving environment.

Lateral displacement is the distance laterally traveled from the start of the projection of the trajectory to its end. The collected data included many scenes of straight-line driving and few turning scenes, such as right/left turns and curves, and there was a concern that training using all the available data would result in over-fitting for scenes representing driving straight ahead. Therefore, we focused on lateral displacement to match the number of images for straight-line and turning situations during training. The threshold of lateral displacement was set to 6 m based on the maximum width of a road where vehicles and pedestrians can coexist, and the labeled images were automatically sorted. In addition, they were manually sorted according to the driving environment based on the following criteria:Roadway: The whole trajectory is on the area that are accessible to cars.Sidewalk: The whole trajectory is on the area that are inaccessible to cars.Roadway and sidewalk: Part of the trajectory is on the roadway and sidewalk.Crosswalk: Part of the trajectory is on the crosswalk.

The number of training data and validation data used for training are shown in parentheses in Table 2. To avoid over-fitting, 3200 training data were extracted so that the number of data with lateral displacement less than 6 m and the number of data with lateral displacement greater than 6 m were uniform, and the number of data in each driving environment was uniform. In addition, 400 validation data were extracted so that the number of data for all conditions was uniform.

### 3.2. Learning Method for Recommended Traversable Area Detection

In this section, we describe learning methods for detecting the appropriate recommended traversable area from a single driving path, focusing on data augmentation and loss weighting. The required functions of the recommended traversable area detection system are as follow:Detect selectable driving directions as recommended areasDo not detect non-recommended driving areas as recommended areasDetect inaccessible directions as non-traversable areasDo not misclassify traversable areas as non-traversable areas

#### 3.2.1. Data Augmentation

The automatically labeled recommended area is the area extending forward from the bottom center of the image, as shown in the fourth column of Figure 5. Therefore, the trained model may over-fit the location and shape of the recommended area in the image. To suppress over-fitting for location and shape, it is considered effective to crop the input image. However, cropping also leads to the loss of context information. Hence, as shown in Figure 8, we propose both lateral crop processing, which functions to preserve lateral context and to suppress over-fitting for longitudinal position, and longitudinal crop processing, which functions to preserve longitudinal context and to suppress over-fitting for lateral position.

#### 3.2.2. Loss Weighting

In this study, we used the cross-entropy loss function and designed loss weighting based on the required functions of the recommended traversable area detection system. Although the automatically labeled image reflects only one run, it may actually run in other directions. Therefore, it is considered effective to reduce the loss weight for the unclassified area relative to the recommended area. However, if only the loss weight for the unclassified area is reduced, the loss weight for the non-traversable area becomes relatively large, and traversable area may be misclassified as non-traversable area. In addition, misclassification of non-traversable area can be caused by the effect of labeled images, including misclassification, as shown in Figure 5(c-1–c-4). Therefore, to prevent misclassification of the non-traversable area, it is effective to reduce the loss weight of the non-traversable area. However, in doing so, it is necessary to understand the effect of reducing the loss weight for the non-traversable area, which may also affect the detection rate of recommended areas. In the verification in Section 4, we determine the output tendency of each class for varying loss weight and propose a loss weighting method for training data, which we used here.

### 3.3. Evaluation Method

In this section, we describe the evaluation dataset used to evaluate the effectiveness of the automatic image labeling method using human driving experience and the learning method described in Section 3.2. In addition, evaluation metrics using the evaluation dataset are illustrated.

#### 3.3.1. Evaluation Dataset

To evaluate the effectiveness of the proposed methods for the required functions of our recommended traversable area detection system, a manually labeled dataset was created for images collected in environments where the roadway and sidewalks coexist. The labeled classes are the nine classes shown in Figure 9. To evaluate the detection rate and misclassification rate of non-traversable areas, we first categorized the image area into two classes according to traversablity. Next, to evaluate whether the trained model fails to detect non-recommended driving areas as recommended areas, we divided the traversable areas into three classes: recommended, non-recommended, and gray, based on traffic rules. Finally, to evaluate the performance in detecting selectable driving directions as recommended areas, we classified the recommended area into six driving directions. The definitions of the three classes of traversable areas, namely recommended, non-recommended, and gray, are shown in Table 3. In this study, we created 100 images labeled into the nine classes shown in Figure 9 and horizontally flipped them to create a dataset of 200 images for evaluation. An example of the labeled images is shown in Figure 10.

#### 3.3.2. Evaluation Metrics

In this section, we describe evaluation metrics used to evaluate each of the four required functions. First, Equation (Equation 3) was used to evaluate the performance in “detecting selectable driving directions as recommended areas” and “not detecting non-recommended driving areas as recommended areas”.
(3)$$\begin{array}{c} R_{i}=\frac{\sum P(p=\mathrm{recommended}\wedge l=i)}{\sum P(l=i)}, \end{array}$$
where i= {straight, left1, left2, right1, right2, other, non-recommended, gray} is the type of manual label, p= {recommended, non-traversable, unclassified} is the prediction label of the output image by the trained model, and l= {straight, left1, left2, right1, right2, other, non-recommended, gray, non-traversable} is the label of the image for evaluation, P(·) indicates the number of pixels per evaluation image satisfying the condition ·, and ∑ means to take the sum of P(·) for each of the 200 evaluation images. As shown in Figure 11a, P(p=recommended∧l=i) represents the intersection area of the area predicted as recommended and the area manually labeled as *i*, and P(l=i) represents the area labeled as *i*. That is, $R_{i}$ is the detection rate of each of the areas labeled with *i*, and the detection rate should be high for the recommended directions, such as straight and left1, and low for the non-recommended directions.

Second, Equation (Equation 4) was used to evaluate the performance in “detecting inaccessible directions as non-traversable areas”.
(4)$$\begin{array}{c} R_{\mathrm{nt}}=\frac{\sum P(p=\mathrm{non}‐\mathrm{traversable}\wedge l=\mathrm{non}‐\mathrm{traversable})}{\sum P(l=\mathrm{non}‐\mathrm{traversable})} \end{array}$$

As shown in Figure 11b, P(p=non‐traversable∧l=non‐traversable) represents the intersection area of the area predicted as non-traversable and the area manually labeled as non-traversable. That is, $R_{\mathrm{nt}}$ is the detection rate of non-traversability, which should be high.

Third, Equation (Equation 5) was used to evaluate the performance in “not misclassifying traversable areas as non-traversable areas”.
(5)$$\begin{array}{c} R_{m}=\frac{\sum \sum_{i}P\left(p=\mathrm{non}‐\mathrm{traversable}\wedge l=i\right)}{\sum \sum_{i}P\left(l=i\right)} \end{array}$$
where $\sum_{i}$ is the sum of P(·) for each label type *i*. This metric represents the misclassification rate of non-traversable relative to traversable, such as recommended and non-recommended, which should be low.

## 4. Experimental Results and Discussion

In this section, we first identify the characteristics of the driving recommendation degree inference process when applying the proposed learning method. Specifically, we discuss the properties of different crop methods and loss weightings and the effectiveness of the automatic labeling method utilizing human experience and knowledge. For verification, training data created using the automatic labeling method, described in Section 3, were used.

### 4.1. Learning Conditions for Baseline

Some of the learning conditions for the baseline were modified based on the conditions shown in Table 1, assuming that the inference is performed at a speed faster than 10 Hz on an in-vehicle PC (Intel Core i7-8750H CPU 2.2 GHz, 32 GB RAM, Geforce RTX 2060 GPU). The changes are described below. We make use of SegNet [8], capable of real-time inference. The image size is 400 × 225, and the batch size is 4 due to memory size constraints during training. A pre-trained VGG-16 [23], available in PyTorch, was used as the encoder. For the training and validation data, automatically labeled images were used, and the number of images is shown in parentheses in Table 2.

### 4.2. Experiments for Characterization by Cropping

In this section, we identify the characteristics of the driving recommendation degree inference for various crop sizes in data augmentation. In cropping, we focus on the effect of the suppression of over-fitting for location and shape of the recommended area and the effect of the size of the range that preserves the context. The training was performed using four different crop sizes: 50 × 225, 400 × 50, 125 × 125, and a combination of 50 × 225 and 400 × 50 (hereafter, denoted as 50 × 225 + 400 × 50). In the condition of 50 × 225 + 400 × 50, training was performed with a half probability of applying either the 50 × 225 crop or the 400 × 50 crop. The length of one side of the cropped image in the 125 × 125 condition is set such that the sum of the image areas used for training is the same as the condition of 50 × 225 + 400 × 50. We compared the 125 × 125 condition and the 50 × 225 + 400 × 50 conditions for their ability to preserve the context in the image. When cropping the image, the position of the left corner was randomly determined so that the area to be cropped fell within the image range, and the area of each size was extracted from that position. Here, the loss weight for each class is set as recommended area:unclassified area:non-traversable area = 1.0:0.01:0.1.

#### 4.2.1. Results

Examples of the inference results are shown in Figure 12. The baseline is the result of training without cropping, and the fourth and subsequent columns are the results of training under the respective crop conditions. In contrast to the baseline, the cropped condition has a different output tendency in the recommended and non-traversable areas. Figure 13 shows quantitatively which areas of the manually labeled image are output as recommended or non-traversable areas, using the evaluation metrics described in Section 3.3.2. With respect to $R_{i}$, however, a larger value is preferable in Figure 13a, and a smaller value is preferable in Figure 13b. Figure 13a shows that the detection rate of recommended areas increases in all crop conditions compared to the baseline. Specifically, left1, left2, right1, and right2 increase by the same amount in the 50 × 225 condition. In the 400 × 50 condition, although left1 and right1 are comparable to the 50 × 225 condition, left2 and right2 increase compared to the 50 × 225 condition. In the 50 × 225 + 400 × 50 condition, the detection rates for straight, left1, left2, right1, and right2 are substantially increased compared to the 50 × 225 condition and 400 × 50 condition, respectively. The 125 × 125 condition has approximately the same detection rate as the 50 × 225 condition. As shown in Figure 13b, although the detection rate of non-recommended areas increases with the crop conditions compared to the baseline, there is no significant difference with the crop size. For the gray in Figure 13b, the same tendency as in Figure 13a is observed. As shown in Figure 13c, although the detection rate of non-traversable areas decreases in the 400 × 50 condition, it is maintained in other crop conditions compared to the baseline. Figure 13d shows that, although the misclassification rate of non-traversable areas is small for all conditions, that of the 50 × 225 + 400 × 50 condition and that of 125 × 125 condition show an increasing trend compared to other conditions.

#### 4.2.2. Discussion

First, we focus on the conditions of 50 × 225 and 400 × 50. As shown in the fourth column of Figure 12, the results of the crop with 50 × 225 show that the recommended area near the vehicle is also present in areas other than the bottom center of the image compared to the baseline, indicating that over-fitting of the lateral position was suppressed. This led to the output of the recommended area not only for the center of the image but also for the left and right directions, which led to an increase in the detection rate of left1, left2, right1, and right2. As a characteristic of the labeled image, non-traversable areas are often included in the upper quarter of the image. Therefore, the preservation of the longitudinal context by cropping longitudinally led to the maintenance of the detection rate of non-traversable areas. For the results of the 400 × 50 crop, the far recommended area is wider than the baseline, as shown in Figure 12(d-5,e-5). This indicates that eliminating the dependence on the longitudinal position prevents over-fitting of the recommended area width on longitudinal position. It is considered that left2 and right2 were increased compared to the 50 × 225 condition by this characteristic. In addition, by cropping in the lateral direction, the continuity of the lateral path is preserved, and the lateral paths in the distant region can be predicted, as shown in Figure 12(d-5). However, by eliminating the dependence on the longitudinal position, the detection rate of non-traversable areas is lower than that of the baseline, as shown in Figure 13c. In summary, the 50 × 225 crop is effective in suppressing over-fitting of the estimated position of the recommended area near the vehicle and maintaining the detection rate of non-traversable areas. However, it has the disadvantage of not being able to predict a continuous lateral path because it breaks continuity. In addition, the 400 × 50 crop is effective in increasing the width of the recommended area and predicting continuous lateral paths; however, it has the disadvantage of reducing the detection rate of non-traversable areas.

Next, we discuss the results for the 50 × 225 + 400 × 50 conditions. The detection rate of recommended area increased the most. In particular, left1 and right1 are substantially increased compared to the 50 × 225 condition and 400 × 50 condition. This is considered to be a result of synergistic effects of suppressing over-fitting of the estimated position of the recommended area near the vehicle and maintaining lateral continuity. In addition, although the detection rate of non-traversable areas was reduced for 400 × 50 cropping alone, 50 × 225 cropping preserved the longitudinal context and maintained the detection rate. However, in addition to the increase in the detection rate of recommended areas, the detection rates of non-recommended and gray areas also increased slightly, as shown in Figure 13b. As for the non-recommended area, as shown in Figure 12(b-6), the roadway in the distant area is predicted to be a recommended area. In addition, as shown in Figure 12(c-6), we found a scene where the recommended area was detected not on the nearby sidewalk, but on the distant roadway. Figure 12(b-2,b-6,c-2,c-6) are enlarged in Figure 14. This may reflect the result of using driving experience on a roadway where the road and the sidewalk are indistinguishable as training data. Comparing Figure 12(e-6) with Figure 12(b-6) and Figure 12(a-6) with Figure 12(c-6), there are differences in the timing at which the sidewalk can be detected as a recommended area in similar scenes. Therefore, it is necessary to verify which direction can be planned as a path when dealing with successive driving recommendation degrees. The reason for the increase in the detection rate of the gray area is that the center of the roadway is also predicted as recommended area on roads with is no distinction between roadway and sidewalk, as shown in Figure 12(a-6).

Finally, we discuss the results for the 125 × 125 condition. Despite the lateral image cropping, the detection rate of non-traversable areas did not decrease with respect to the baseline results as it did in the 400 × 50 condition. This is because the length of one side of the cropped image is more than half the height of the image, which is long enough to capture the context of the non-traversable areas in the upper quarter of the image. In terms of the recommended area, it increases compared to the baseline. This is likely due to the suppression of over-fitting of longitudinal and lateral positions. However, the detection rate of the recommended area is lower than that of the 50 × 225 + 400 × 50 conditions. This is likely due to the lack of information on the lateral continuity of the paths in the image after cropping. As a result, there is a break in the lateral path, as shown in Figure 12(a-7,d-7). In other words, the inability to maintain path continuity information by cropping in both the longitudinal and lateral directions to the same extent may lead to a decrease in the detection rate of recommended areas.

From the above, we find that it is possible to prevent over-fitting of the position, while maintaining context by combining longitudinally and laterally cropped images under the condition that the sum of the area of the used images for training is equal, and, as a result, it is confirmed that the recommended area detection performance using the automatically labeled images based on a single driving path can be improved.

### 4.3. Experiments for Characterization by Loss Weighting

In this section, we characterize the properties of varying loss weight for unclassified and non-traversable areas. Specifically, as described in Section 3.2.2, it is considered effective in satisfying the required functions if the loss weight for the unclassified area and that for the non-traversable area are both small compared to the loss weight for the recommended area. Therefore, the training was performed under the following conditions: the loss weight for each class is set as recommended area:unclassified area:non-traversable area = 1.0:*u*:0.01 or recommended area:unclassified area:non-traversable area = 1.0:0.01:*v*, where u=v={0.1,0.01,0.001}. As a data augmentation, we use a method of applying the 50 × 225 + 400 × 50 condition from the previous section.

In addition, the features of the labeled images used in this study are summarized below as necessary information for analysis of the results.

Recommended areas and unclassified areas are often adjacent to each other.Unclassified areas and non-traversable areas are often adjacent to each other.Recommended areas and non-traversable areas are rarely adjacent to each other.

#### 4.3.1. Results

The trend of each metric for varying loss weight for unclassified areas is shown in Figure 15 and that for non-traversable areas is shown in Figure 16. Here, the detection rate of recommended areas is expressed as $R_{\mathrm{recommended}}$ collectively, without distinguishing classes such as straight and left1, to understand the trend in the entire recommended area. Figure 15 shows that, as *u* decreases, the detection rate of recommended areas, the detection rate of non-recommended areas, the detection rate of non-traversable areas, and the misclassification rate of non-traversable areas all show an increasing trend. In addition, Figure 16 shows that, as *v* decreases, the detection rate of recommended areas, the detection rate of non-traversable areas, and the misclassification rate of non-traversable areas show a decreasing trend. However, as shown in Figure 16b, the detection rate of non-recommended and gray areas did not show a decreasing trend with increasing or decreasing *v*.

#### 4.3.2. Discussion

First, we analyze the cause of the trend shown by each evaluation metric when the loss weight is changed. The increasing trend of each metric shown in Figure 15 is due to the fact that reducing the loss weight for the unclassified area facilitates the output of recommended and non-traversable areas because the loss weight for the recommended area and the non-traversable area become relatively larger. The decreasing trend of metrics shown in Figure 16a,c,d are due to the fact that reducing the loss weight for the non-traversable area facilitates the output of unclassified areas, which are often adjacent to non-traversable areas, and obstructs the output of recommended areas, which are often adjacent to unclassified areas. Nevertheless, the lack of a decreasing trend in the metric shown in Figure 16b is due to the fact that reducing the loss weight for the non-traversable area facilitates the output of recommended areas in the area adjacent to non-traversable areas, facilitating the output of recommended areas in non-recommended areas, which are often adjacent to non-traversable areas, as shown in Figure 10.

Next, we describe the design policy of the loss weights based on the results obtained in Section 4.3.1. The relationships between the detection rate of recommended areas and the detection rate of non-recommended areas and between the detection rate of recommended areas and the misclassification rate of non-traversable areas when the loss weights are changed are schematically shown in Figure 17. The relationship between the detection rate of recommended and non-recommended areas can be adjusted by the loss weight for the unclassified area, as shown in Figure 17a. In addition, the relationship between the detection rate of recommended areas and the misclassification rate of non-traversable areas is a trade-off, as shown in Figure 17b, and can be adjusted by both the loss weight for the unclassified area and that for the non-traversable area. However, when reducing the misclassification rate of non-traversable areas, it is desirable to minimize the reduction in the detection rate of recommended areas, and this means that the slope of a line shown in Figure 17b is small, as indicated by the red dotted line. The slope of the line indicates the ratio of the change in the detection rate of recommended areas to the change in the misclassification rate of non-traversable areas, and it is defined as the sensitivity ratio (SR).

Figure 18a,b shows graphs with the logarithm of the loss weights on the lateral axis and the longitudinal axis as the recommended area detection rate $R_{\mathrm{recommended}}$ shown in Figure 15a and Figure 16a, respectively, and Figure 18c,d, shows graphs with the longitudinal axis as the non-traversable area misclassification rate $R_{m}$ shown in Figure 15d and Figure 16d, respectively. Each graph in Figure 18 shows an approximate line, and the ratio $SR_{\log k}$(k={u,v}) of the slope of the line for each weight is calculated from Equation (Equation 6).
(6)$$\begin{array}{c} SR_{\log k}=\frac{\frac{\Delta R_{\mathrm{recommended}}}{\Delta \log k}}{\frac{\Delta R_{m}}{\Delta \log k}}=\frac{\Delta R_{\mathrm{recommended}}}{\Delta R_{m}}=SR. \end{array}$$

Table 4 lists the slope of the approximate line shown in Figure 18 and the ratio of the slope of the approximate line obtained from Equation (Equation 6). As shown in Table 4, the ratios of the slopes of the lines for the weights are $SR_{\log u}=21.3$ and $SR_{\log v}=4.44$, respectively, and it is confirmed that the weight *v* for the non-traversable area has a smaller SR for the weight change. Therefore, it is desirable to change the loss weight for the non-traversable area to reduce the misclassification rate of non-traversable areas, while suppressing the reduction of the detection rate of recommended areas.

Based on these results, the design policy of loss weights for the unclassified area and the non-traversable area when learning with training data used in this study is as follows.

The loss weight for the unclassified area is first set based on the trade-off between the detection rate of recommended and non-recommended areas.The loss weight for the non-traversable area is then adjusted based on the trade-off between the detection rate of recommended areas and the misclassification rate of non-traversable areas.

Examples of the inference results of the trained model after loss weight adjustment according to this design policy are shown in Figure 19. We can confirm that it is possible to output recommended areas in directions other than the roadway as well as manually labeled images, and that the area can be enlarged beyond what can be projected from a single run. This result shows that the weakly-supervised semantic segmentation approach using training data based on human-selected paths proposed in this paper is useful for driving recommendation degree inference in environments where features such as physical features and edges cannot be measured.

### 4.4. Limitations

In this study, the effectiveness of the proposed learning method was verified under the constraint that the images shown in parentheses in Table 2 were used as training data and the dataset described in Section 3.3.1 was used for evaluation. The scenes in these training and evaluation datasets are somewhat uniform in terms of weather conditions and driving environments. Therefore, the effect of the reflection of non-traversable objects in the puddles, as shown in Figure 20, and the shadows of lattice shapes with features similar to “fence”, one of the CityScapes classes, may increase the misclassification of non-traversable areas. In response to this issue, Barnes et al. [9] showed that good segmentation results were obtained even under rainy and sunny conditions using data acquired under various weather conditions as training data. Since the training data used in this study were also generated automatically, it is easy to obtain training data under different conditions. In the future, it is necessary to identify situations where misclassification of non-traversable areas occurs and to verify the effectiveness of proposed methods when using labeled images of such scenes for training. Moreover, the effects of overexposure or underexposure can be addressed by using a high-dynamic-range camera [9]. However, it may also be necessary to evaluate the impact of the degree of whiteness or blackness in the image on the inference and to determine whether the image can be used for inference during online processing. In addition, as a result of verification using still images, it was confirmed that there were situations where the recommended area was detected on the actual non-recommended area, and there was a risk of planning a path on the non-recommended area. Another possibility is that, although the misclassification rate of non-traversable areas is small, as shown in Figure 16d, it may affect path planning. Therefore, it is necessary to understand the effect on path planning when using a continuous driving recommendation degree and to evaluate the usefulness of the recommended traversable area detection system, including path planning.

## 5. Conclusions

In this paper, we propose a recommended traversable area detection system for topological map-based navigation that can be adapted to environments with no edges using weakly supervised semantic segmentation. First, we developed a method for automatic labeling of the recommended area using driving data with a personal mobility vehicle to generate training data for semantic segmentation.

Second, we focused on data augmentation and loss weighting for each class to detect multiple recommended driving directions based on a single path and to characterize the performance of the recommended area detection. When using automatically labeled images, we found that cropping can prevent over-fitting for position and shape, but that it is also important to preserve the context when cropping. It was found that cropping only in the lateral direction was effective in suppressing over-fitting for the longitudinal direction while maintaining the context in the lateral direction, and cropping only in the longitudinal direction was effective in suppressing over-fitting for the lateral direction while maintaining the context in the longitudinal direction. By applying both cropping methods, each effect was reflected synergistically, and this approach was found to be effective in improving the recommended area detection performance. For the loss weights, we analyzed the characteristics when the weights for the unclassified area and the non-traversable area were changed, and the weights of the loss function were designed by focusing on the sensitivity of the detection rate of the recommended area to the change of each weight. As a result of applying the adjusted weights, it was found that weakly-supervised semantic segmentation using training data based on a human-selected path is useful for driving recommendation degree inference in environments where features such as physical features and edges cannot be measured.

Since the data described in this study were used to verify the effectiveness, the effectiveness of the proposed learning method when using data acquired in different weather conditions and driving environments will be verified in the future. In addition, the validation was conducted using only still images; however, the driving recommendation degree inference results change sequentially even in similar scenes. Therefore, we plan to verify the usefulness of the proposed recommended traversable area detection system, including path planning, using continuous recommendation degree information.

## Figures and Tables

**Figure 1 sensors-21-00437-f001:**
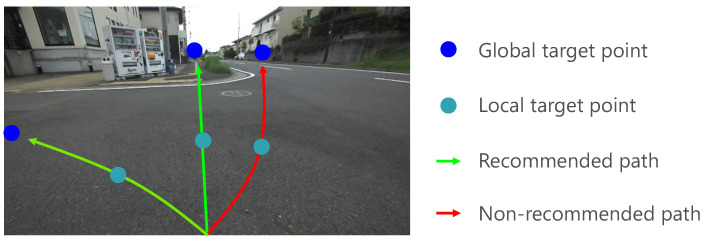
Potential path prediction.

**Figure 2 sensors-21-00437-f002:**
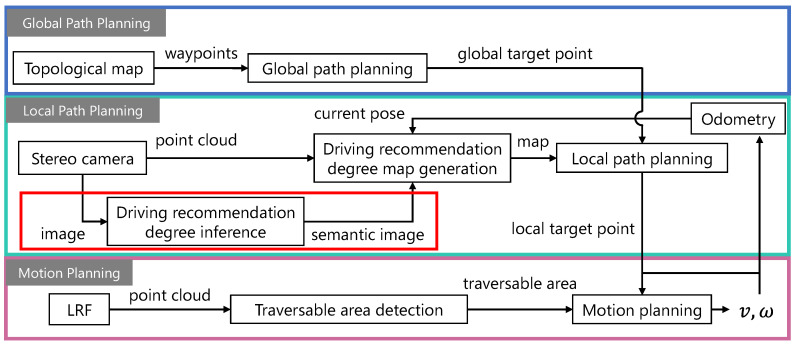
Diagram for overall navigation system using topological map and driving recommendation degree map.

**Figure 3 sensors-21-00437-f003:**
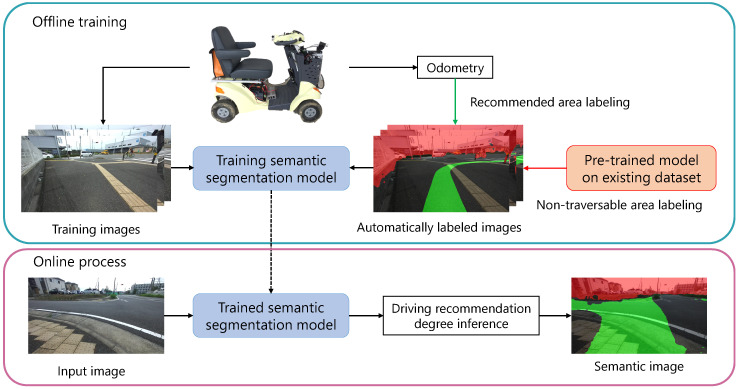
Pipeline of driving recommendation degree inference via offline training using automatically labeled images and online semantic segmentation.

**Figure 4 sensors-21-00437-f004:**
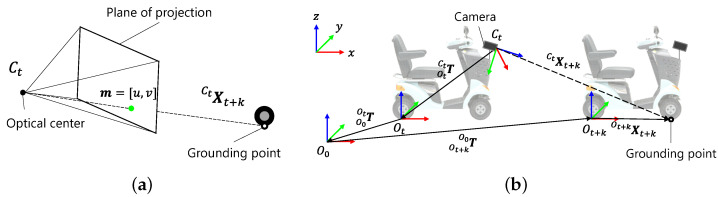
Definitions of geometric relationships: (**a**) perspective projection of wheel grounding point; and (**b**) coordinate origin and coordinate transformation for deriving ${\;}^{C_{t}}X_{t+k}$.

**Figure 5 sensors-21-00437-f005:**
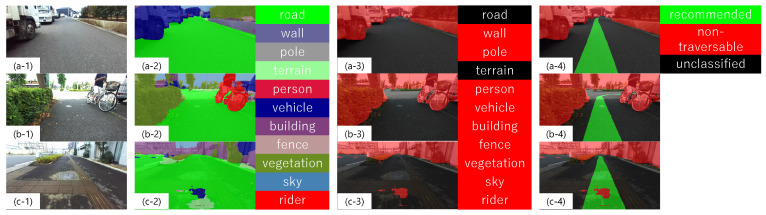
Example of non-traversable area labeling: Column 1, original image; Column 2, labeled image in 11 classes and colors to indicate each class; Column 3, labeled image of a class of non-traversable objects (road, wall, pole, person, vehicle, building, fence, vegetation, sky, and rider); and Column 4, labeled image of recommended area and non-traversable area. (**a-1**)–(**c-4**) are the image numbers referred to in the text.

**Figure 6 sensors-21-00437-f006:**
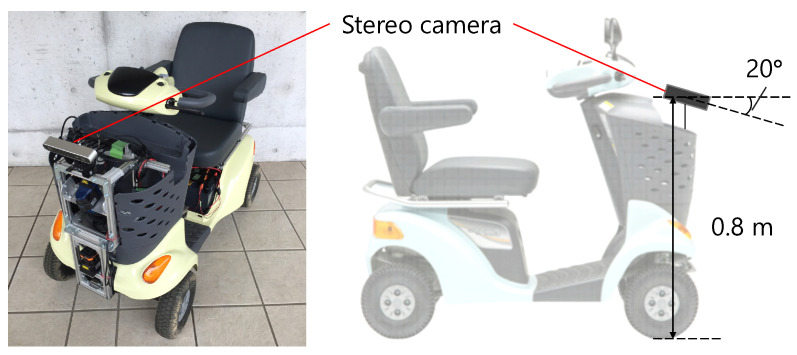
Experimental vehicle and sensor mounting position.

**Figure 7 sensors-21-00437-f007:**
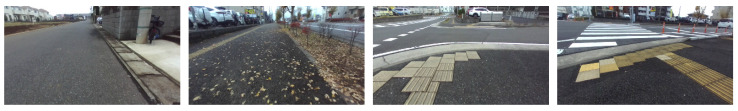
Representative snapshots of driving environments in Japan.

**Figure 8 sensors-21-00437-f008:**
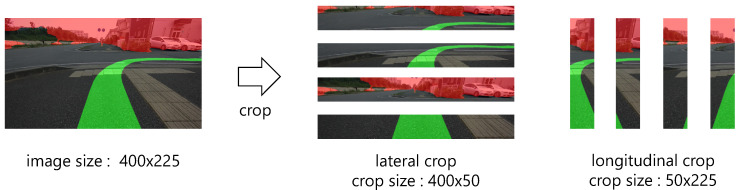
Example of image cropping to prevent over-fitting for position and shape of recommended area (green area).

**Figure 9 sensors-21-00437-f009:**
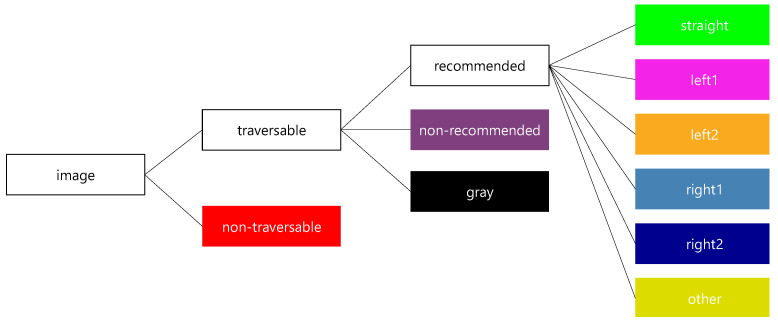
Classes and colors for labeling. The label “straight” means the area indicating the direction to go along the road, “left1” (“right1”) means an area indicating left or right turn direction before crossing the road, “left2” (“right2”) means an area indicating left or right turn direction after crossing the road, and “other” means an area that does not determine the direction but is recommended.

**Figure 10 sensors-21-00437-f010:**
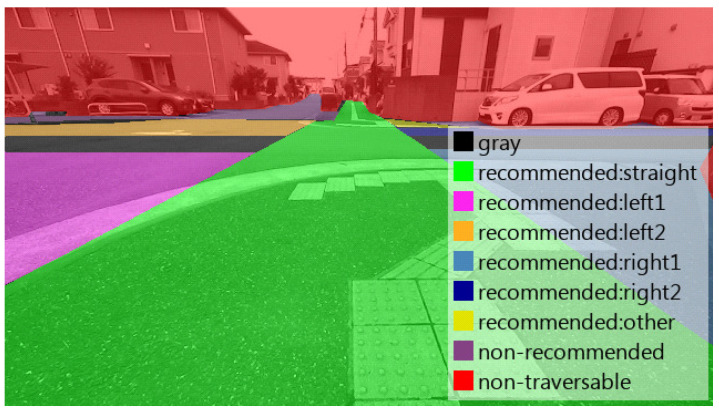
Example of labeled image for evaluation.

**Figure 11 sensors-21-00437-f011:**
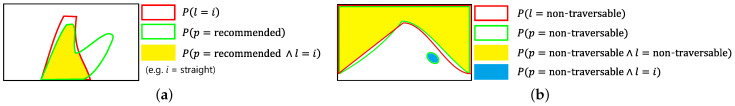
Image of evaluation area. (**a**) Evaluation area for $R_{i}$. (**b**) Evaluation area for $R_{\mathrm{nt}}$ and $R_{m}$.

**Figure 12 sensors-21-00437-f012:**
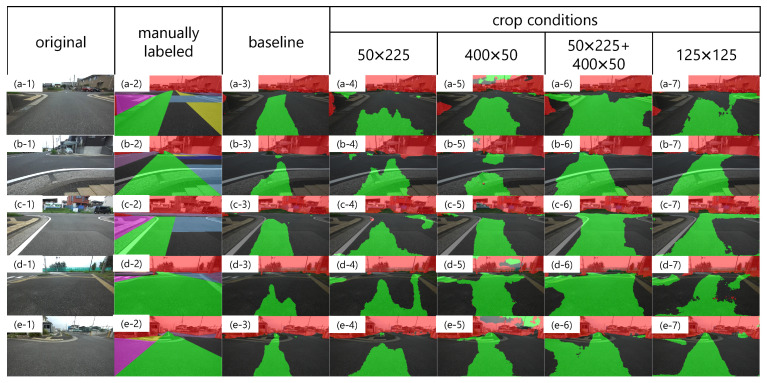
Qualitative comparisons between different crop sizes. (**a-1**)–(**e-7**) are the image numbers referred to in the text.

**Figure 13 sensors-21-00437-f013:**
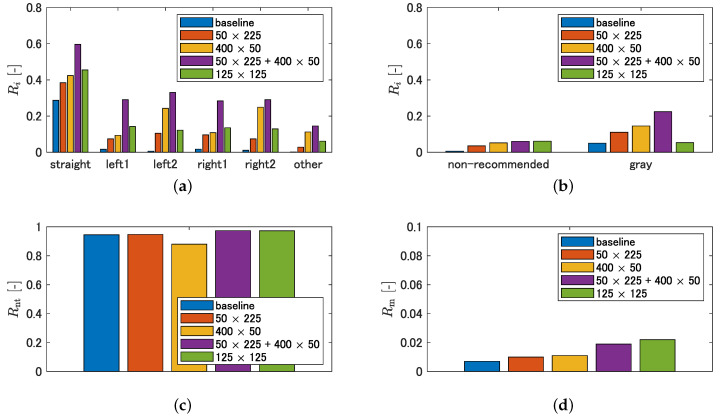
Quantitative evaluations of effectiveness of cropping: (**a**) detection rate of recommended area; (**b**) detection rate of non-recommended area and gray area; (**c**) detection rate of non-traversable area; and (**d**) misclassification rate of non-traversable area.

**Figure 14 sensors-21-00437-f014:**
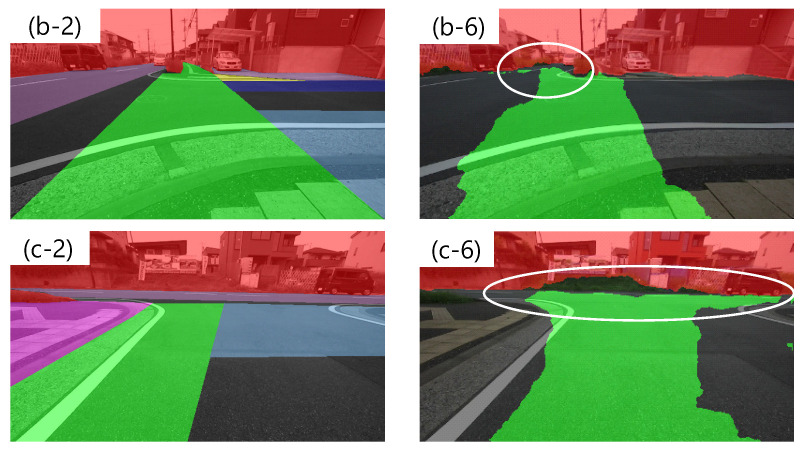
Examples of outputting recommended areas on non-recommended areas. (**b-2**)–(**c-6**) are enlarged images of Figure 12(b-2,b-6,c-2,c-6).

**Figure 15 sensors-21-00437-f015:**
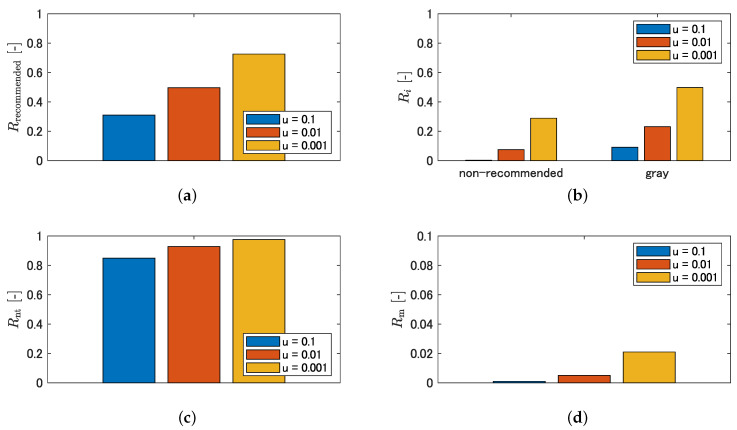
Quantitative characterization by varying loss weight for unclassified area: (**a**) detection rate of recommended area; (**b**) detection rate of non-recommended area and gray area; (**c**) detection rate of non-traversable area; and (**d**) misclassification rate of the non-traversable area.

**Figure 16 sensors-21-00437-f016:**
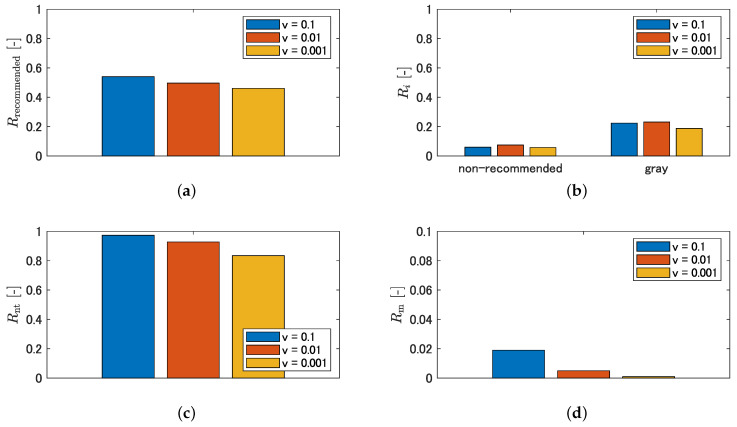
Quantitative characterization by varying loss weight for non-traversable area; (**a**) detection rate of recommended area; (**b**) detection rate of non-recommended area and gray area; (**c**) detection rate of non-traversable area; and (**d**) misclassification rate of non-traversable area.

**Figure 17 sensors-21-00437-f017:**
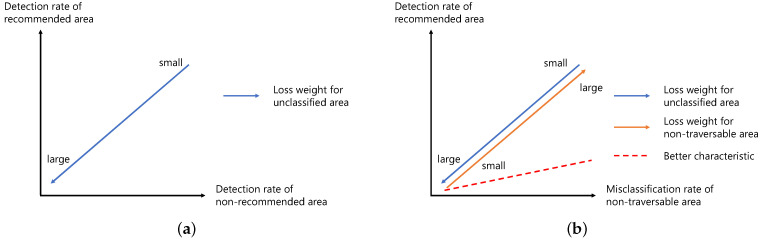
Trade-off by loss weighting: (**a**) trade-off between detection rate of non-recommended area and detection rate of recommended area; and (**b**) trade-off between misclassification rate of non-traversable area and detection rate of recommended area.

**Figure 18 sensors-21-00437-f018:**
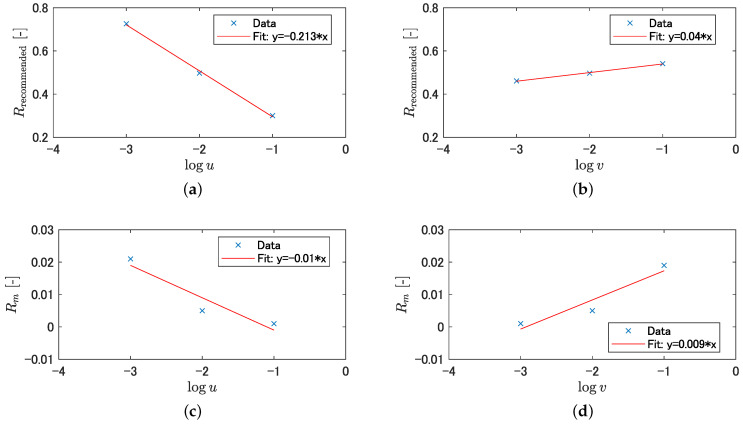
Changes in the detection rate of recommended area and the misclassification rate of non-traversable area by loss weighting: (**a**) changes in the detection rate of recommended area by varying loss weight for unclassified area; (**b**) changes in the detection rate of recommended area by varying loss weight for non-traversable area; (**c**) changes in the misclassification rate of non-traversable area by varying loss weight for unclassified area; and (**d**) changes in the misclassification rate of non-traversable area by varying loss weight for non-traversable area.

**Figure 19 sensors-21-00437-f019:**
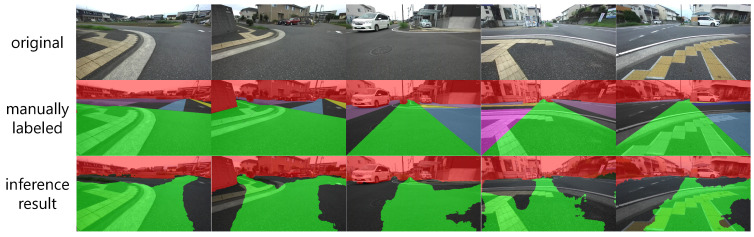
Inference results of trained model after loss weight adjustment with 50 × 225 + 400 × 50 cropping.

**Figure 20 sensors-21-00437-f020:**
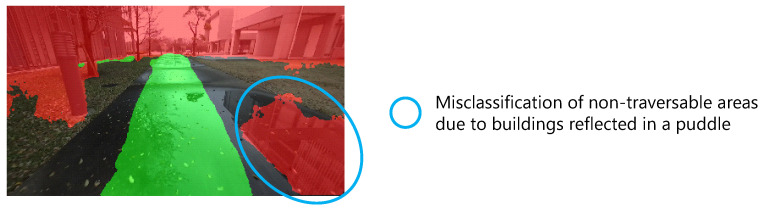
Example of misclassification of non-traversable areas due to non-traversable objects reflected in a puddle.

**Table 1 sensors-21-00437-t001:** Leaning conditions.

Conditions		Value
learning rate (lr)	base_lr [20]	0.01
	power [20]	0.9
	decoder_lr/encoder_lr	0.1
optimizer	momentum	0.9
	weight decay	0.0001
data loader process	image size	1024 × 512
	batch size	2
	max epoch	200
GPU (NVIDIA Geforce RTX 2060)	memory size (GB)	6

**Table 2 sensors-21-00437-t002:** The number of automatically labeled images including horizontally flipped images. The numbers in parentheses are the number of images used for training and validation, respectively, which were generated using data collected in different environments.

	Lateral Displacement (ld)	Roadway	Sidewalk	Roadway and Sidewalk	Crosswalk	
training	ld < 6	11,210 (500)	9144 (500)	1116 (300)	926 (300)	
data	ld ≥ 6	994 (300)	938 (300)	838 (500)	1034 (500)	
validation	ld < 6	7466 (50)	1150 (50)	604 (50)	150 (50)	
data	ld ≥ 6	306 (50)	72 (50)	298 (50)	184(50)	

**Table 3 sensors-21-00437-t003:** Definition of manual labeling for evaluation dataset.

Class	Definition
recommended	sidewalk of a road with a distinction between a roadway and a sidewalk roadside of a road with no distinction between a roadway and a sidewalk crossing area of a road without a center line crosswalk
non-recommended	roadway of a road with a distinction between a roadway and a sidewalk private area such as parking lots planting on a sidewalk
gray	crossing area of a road with a center line roadside strip of a roadway with a distinction between a roadway and a sidewalk road center of a road with no distinction between a roadway and a sidewalk

**Table 4 sensors-21-00437-t004:** Derivation of sensitivity ratio (SR).

	$\frac{\Delta R_{\mathrm{recommended}}}{\Delta \log k}$	$\frac{\Delta R_{m}}{\Delta \log k}$	${\mathrm{SR}}_{\log k}$	
k=u	−0.213	−0.01	21.3	
k=v	0.04	0.009	4.44	

## Data Availability

The data presented in this study are available on reasonable request from the corresponding author. The data are not publicly available due to privacy.

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
