# Peer review of "Weakly-Supervised Recommended Traversable Area Segmentation Using Automatically Labeled Images for Autonomous Driving in Pedestrian Environment with No Edges"

_sensors, 2021, doi:10.3390/s21020437_

Round 1

Reviewer 1 Report

This manuscript presents a method for the detection of a recommended traversable area for the autonomous personal mobility systems in environments with no edges and distinctions between the roadway and the sidewalk. The method uses a weakly-supervised semantic segmentation based on paths selected by humans and images taken by a camera on the mobility system. Additionally, a data augmentation method and a loss weighting method for detecting the recommended traversable area from a single human-selected path are proposed.

Comments:

The manuscript is very well written and easily readable. The processing of the obtained datasets and the applied methods are described in detail. The results are clearly presented and followed by adequate discussion and conclusions accompanied by the challenges of the study and plans for future research.

I believe the manuscript is acceptable for publication in Sensors journal. Before that, I have one minor comment that I would like the authors to address:

  1. Please extend section 2. Related works with the discussion of several more recent and relevant references, as you have only discussed three references in this section.
  2. Refer also to MDPI Sensors papers in this field. 

Reviewer 2 Report

The paper deals with an interesting topic autonoumous driving with a specific focus on visual detection. The paper is well described from a methodological point of view. Some more issues should be introduced in term of experimental valiudation, also in term of experimental settings details in order to provide to the paper a more reliable behaviour from a scientific point of view 

Reviewer 3 Report

The authors have presented an excellent and well-written paper. It is clear their effort in developing and writing this research.

I have just a few suggestions.
I think the title is misleading and focuses on the final application. However, the paper has presented much more than that. It is a framework for automatic labeling and fusing information applied to the traversable area problem. My opinion is that there is almost enough material for two papers, one regarding the framework and another one the application.

A concern that I particularly have when reading and revising papers for autonomous decision-making systems is the lack of discussing their operational limits. For instance, figure 16 - d shows vert low misclassification rates. However, a more in-depth discussion on what decision should be taken should be done.

Another minor concern is how the system performs in different weather conditions.

For instance:
- after raining when the streets are wet and may mirror the sensor's image.
- on a sunny day with shadows and infrared reflections that reduces the ZED reliability.
